# A Systematic Review of Factors Associated with Sport Participation among Adolescent Females

**DOI:** 10.3390/ijerph19063353

**Published:** 2022-03-12

**Authors:** Casey S. Hopkins, Chris Hopkins, Samantha Kanny, Amanda Watson

**Affiliations:** 1School of Nursing, Clemson University, Clemson, SC 29634, USA; anwtsn@g.clemson.edu; 2Department of Health Sciences, Furman University, Greenville, SC 29613, USA; chris.hopkins@furman.edu; 3Department of Public Health Sciences, Clemson University, Clemson, SC 29634, USA; skanny@g.clemson.edu

**Keywords:** sport participation, adolescents, females, sport dropout, theory of planned behavior, physical activity

## Abstract

Sport participation provides a direct means to attain health-enhancing physical activity; however, sport participation declines during adolescence, and over 85% of adolescent females fail to meet the recommended 60 min of moderate-vigorous physical activity daily. Given the importance of overcoming barriers to sport and increasing equity in women’s sports, the purpose of this systematic review was to identify factors associated with sport participation among adolescent girls and operationalize those factors into theoretical constructs to guide future research. Six databases were systematically searched, and 36 records were included for review. Factors impacting girls’ sport participation were categorized as personal, peer, family, socioeconomic, environmental, or other factors. Of these categories, personal factors, including self perceptions and desirable personal outcomes related to sport, were most frequently associated with sport participation. Most research on girls’ sport participation lacks theoretical framework, so to aid future studies, this review categorized important participatory factors into the constructs of the theory of planned behavior. Future research would benefit from theory-driven prospective approaches to make clear and consistent predictions about factors impacting sport participation, as well as mixed-method approaches aimed to provide more robust understanding of girls’ experiences with and perceptions of factors impacting their participation in sports.

## 1. Introduction

Adolescence is a transformational time of life between childhood and adulthood, from around ages 10 to 19 years, when rapid physical, cognitive, and psychosocial growth takes place. Laying the groundwork for healthy behaviors is paramount during these formative years [1]. Adolescents who are active in sports have better outcomes across the lifespan with regard to physical, mental, social, and emotional health [2]. Sports encourage physical activity among adolescents by providing a structured environment where expectations are defined and goals are shared between youth and involved adults, namely coaches and parents [3]. The health benefits of physical activity gained through sports are extensive, from improving emotional health through fostering social relationships [4], encouraging resilience and self esteem [5], positively impacting academic performance [6], to decreasing risks of physical disease and disability [7]. Despite decades of data to support the benefits of physical activity through sport participation and efforts from global public health organizations to encourage physical activity, involvement in sports continues to decline at the greatest rate during adolescence, with the average adolescent dropping out of sports at 11 years of age [8].

According to the World Health Organization, adolescents should engage in 60 min of moderate to vigorous exercise daily; but sadly, 81% of adolescents aged 11–17 are not sufficiently physically active [1,9]. While adolescence has been identified as a prominent time for sports dropout in both males and females [10,11,12], adolescent girls are at a greater risk of dropping out of sports at an earlier age than their male counterparts [10,11,13]. Globally, 85% of female adolescents are not meeting recommended levels of physical activity [1]. Reviewing the factors contributing to female adolescents’ sport involvement is important, so that relevant stakeholders, including public health officials, educators, coaches, parents, and healthcare providers, may be able to provide resources to enhance the experience of female adolescent athletes and promote positive atmosphere for regular physical activity. The most recent Women’s Sports Foundation (2020) publication on equity in women’s sports called for research to identify factors impacting sport participation among girls for the purpose of overcoming barriers to sport and physical activity in communities [14].

Public health researchers and practitioners often employ theories to deepen our understanding of health behaviors and suggest ways to promote positive health behavior change [15]. Despite the success of employing behavioral theories to better understand many health-related behaviors, researchers have seldomly used theory-guided approaches to understand adolescent girls’ intentions to participate in sports, despite sports being a common means to achieve health-enhancing physical activity within this physically inactive population. An increased emphasis on employing theoretical frameworks in research on girls’ sport participation may lead to a better understanding of important factors that facilitate sport participation in this population.

### 1.1. Theory of Planned Behavior

The theory of planned behavior has been widely applied to understand and predict human behavior, including adolescents’ intentions to engage in physical activity [16,17,18,19]. Thus, the theory of planned behavior can be used as a framework to better understand adolescent girls’ intentions of participating in sports.

The theory of planned behavior posits three constructs that predict *intention* to engage in behaviors: *attitudes*, *subjective norms*, and *perceived behavioral control* [16]. *Attitudes* are defined as the beliefs one has about behaviors, such as whether engaging in a certain behavior will result in a desirable outcome [20,21,22]. For example, an adolescent girl may think “Practicing soccer gives me exercise which keeps me healthy and that is good.” *Subjective norms*, or an individual’s perception of whether other people believe they should perform a behavior, can be divided into two categories: injunctive and descriptive normative beliefs [23]. Injunctive norms are the perception that a particular group will approve or disapprove of a behavior [24,25]. For example, “My parents think it is good for me to play soccer.” Descriptive norms are informed by an individual’s perception of what is being done by others [23]. For example, “Most of the other girls in my class play soccer.” Therefore, subjective norms may impact one’s intentions through their perceptions of what ought to be (injunctive norms) and what is (descriptive norms). Finally, *perceived behavior control* is defined as an individual’s belief that they will be able to perform the behavior considering the factors that either facilitate or impede on partaking in that behavior [26]. For example, an adolescent girl may consider all the factors in her life that may impact her involvement in sport and conclude, “Playing soccer over the next three months will be easy for me.”

### 1.2. The Present Study

The aims of this review are to (1) collect and synthesize findings from a broad sample of relevant literature to examine factors that influence adolescent girls’ participation in sports and (2) categorize each of the influential factors into constructs of the theory of planned behavior to guide future research and practice promoting adolescent girls’ sport participation.

## 2. Methods

### 2.1. Data Sources

The following databases were searched in October 2020 to conduct this review: PubMed, CINAHL, Academic Search Complete, SportsDISCUS, APA PsychInfo, and Medline. The search strategy aimed to find articles examining factors associated with sport participation and attrition among adolescent girls. Search terms included: “Girls” or “females” or “adolescent girls” or “female adolescents”, and “sport participation” or “sport involvement” or “sport attrition” or “sport dropout.” There were no limits placed on the date of publication for the search, so that we could evaluate the current state of science and how the science has progressed over the years. Search records were saved in Refworks. The search strategy was also saved to be able to recreate the results of the review.

### 2.2. Inclusion/Exclusion Criteria

Articles were considered for inclusion if they were peer-reviewed, original research articles addressing factors associated with sport participation or attrition in female adolescents and were published in English. Articles were excluded if (1) the focus was on individuals with intellectual or developmental disabilities and did not address adolescents with normal development, and (2) there was no stratification for differences between sex if the study included males and females.

### 2.3. Search Process

A total of 2899 records were identified with the initial search of the databases using the terms above. After removing duplicates, there were 2158 records. Records were excluded if they were not published in English, did not address an adolescent population, and were not empirical research articles. To filter records for the adolescent population we set a search parameter within the databases to include records studying subjects under the age of 17. Following these exclusions, 517 records remained. Screening of titles and/or abstracts yielded the exclusion of 406 records that were not relevant to the objective of the review. The remaining 111 records were reviewed in entirety for content. Records were excluded if sport participation or attrition was not a dependent variable of the study, the findings were not stratified for gender, the population was largely made up of adults (over age 17) or children (under age 10), and they did not stratify for age. Finally, there were 36 remaining records addressing factors associated with female adolescent sport participation that were included in this systematic review. A flow diagram based on recommendations from The PRISMA Group for reporting items for systematic reviews illustrating the record selection process is provided in Figure 1 [27].

### 2.4. Quality Appraisal

Data quality of the 36 records was assessed by each author individually using the Joanna Briggs Institute Checklist for Critical Appraisal of Analytical Cross-Sectional Studies and Critical Appraisal of Cohort Studies [28]. The records were assessed for methodological quality, trustworthiness, and relevance. Additionally, the level of evidence for each record was assessed by the authors using the Johns Hopkins Nursing Evidence-Based Practice Research Evidence Appraisal Tool in order to determine the strength and quality of each record [29]. The critical appraisal and level of evidence for each record is provided in Appendix A. The authors collaboratively reviewed findings from the quality appraisals of each record. Based on the level of evidence assigned to each record, 21 were considered high quality, indicating they produced consistent, generalizable results with definitive conclusions, adequate controls, and sufficient sample size. The remaining 15 records were considered to be good quality, indicating they produced reasonably consistent results with fairly definitive conclusions, some control, and had a sufficient sample size. If records were scored as low quality, they would have been excluded from the review. Consensus was reached among the authors, and the decision was made to move forward with the review of all 36 records.

## 3. Results

The 36 records reviewed consisted of various study designs, all of which were quantitative. Small studies using cross-sectional surveys were reviewed, as well as studies conducting secondary analyses of large data sets. Included records ranged in years of publication from 1976 through 2020, with 61% of records having been published since 2010. Record origination spanned the globe, representing countries in Europe, Asia, north and south America, Africa, and Australia. A total of 25 studies included both male and female participants, and 11 studies included only females. The ages of participants ranged from 5 to 21 years. These studies stratified for sex and age; therefore, we were able to compile a list of factors impacting sport participation among female adolescents. A descriptive comparison of study characteristics is provided in Table 1.

The first aim of this review was to examine the factors influencing girls’ participation in sports. Thus, among the articles included in this review, any variable associated with girls’ sport participation was recorded. These variables were then grouped into categories of the following contributing factors that impacted sport participation: personal, peer, family, socioeconomic, environmental, and other factors. A comprehensive list of the contributing factors, variables, and the study in which each variable was identified is included in Table 2. Personal factors were observed to impact sport participation most frequently, and these were often related to desirable outcomes related to sport (i.e., enjoyment, skill development, and fitness) and perceptions of self. Variables related to “self” varied from issues of physicality and behaviors to psychological issues, such as self worth and self determination. Family factors were the second most common impactful variables influencing sport participation among girls. Characteristics of parents, from their employment status to their level of physical activity, along with the support offered from the family, play a major role in whether girls participate in sports. Next, the number of biological (age, BMI, height) and peer-related factors were equally identified in this review as impacting sport participation among girls. These factors were followed by socioeconomic and environmental factors, which often related to limited resources or accessibility to participate in sports. Lastly, factors specific to the type of sport or coaching influences were also found to impact sport participation.

Of the 36 records included in this review, only 7 were guided by theory. In order to promote the use of behavioral theories in future research on girls’ sport participation, the second aim of this review was to categorize influential variables into the constructs of the theory of planned behavior. All three constructs (*attitudes*, *subjective norms*, and *perceived behavioral control*) were represented in the included studies [16]. Some variables did not align with the constructs of the theory of planned behavior, therefore, we conceptualized them as peripheral to the theory and depicted them as *other factors* with the potential to impact attitudes, subjective norms, and perceived behavioral control. Figure 2 displays the categorization of influential variables into the theory of planned behavior.

### 3.1. Attitudes

Nine of the variables were included in the attitude construct, as they were determined by girls’ enjoyment of participating in sports and the potential outcomes of sport participation, such as gaining skills, fitness, opportunities for challenge or competition, friendships, community (being on a team), and better mental health.

### 3.2. Subjective Norms

Twelve variables were operationalized into the subjective norms construct, with six of them being related to family, namely parental factors. These variables included both injunctive normative beliefs (parent and peer encouragement) and descriptive normative beliefs (parental exercise and peer acceptance). Co-ed physical education was a factor included as a subjective norm because the premise of this study variable dealt with peer influence pertaining to perceived gender roles, an injunctive normative belief [30].

### 3.3. Perceived Behavior Control

Nineteen of the variables were included in the construct of perceived behavioral control, as they were determined to either facilitate or impede girls’ control in participating in sports. Mental health was included in perceived behavioral control and attitudes because of the bidirectional association between sport involvement and mental health that was found by Vella et al. [31]. An adolescent girl may believe that participating in sports is desirable because it will improve her mental health, or she may feel less perceived control over her ability to participate in sports due to her mental health status. Ten of the variables categorized as perceived behavioral control were personal factors of self worth, esteem, determination, competence, and other perceptions of self that could facilitate or impede an individual’s perceived control to participate in sports. The other eight variables were either directly or indirectly related to socioeconomic status, including factors related to parental employment, perceived family wealth, access to facilities and equipment, opportunities to play, sport club membership, and physical location of the girls’ residence.

### 3.4. Other Factors

Fifteen variables were identified as being peripheral to the theory. These factors could influence attitudes, subjective norms, and/or perceived behavioral control but were not able to be clearly operationalized into the constructs of theory of planned behavior. The *other factors* were largely related to biological or physiological variables, such as age, maximum oxygen uptake, BMI, height, whether the child was breastfed or not, or had ever sustained an injury. Decisions to engage in dieting and substance use may influence the constructs predicting girls’ intentions and, thus, were included here. Traditional family structure, parents’ immigration status, parents’ education level, and having experienced verbal abuse from a coach were also included as being factors likely to impact the attitudes, subjective norms, and perceived behavioral control of girls. Lastly, one’s personal investment, the type of sport played, and the frequency of involvement were peripheral to the theory and may impact the constructs that predict intention.

## 4. Discussion

The present study aimed to systematically review relevant literature to determine important factors related to girls’ sport participation and categorize these factors into constructs of the theory of planned behavior. The results of this systematic review revealed personal factors that related to self perceptions, and desirable outcomes of sport were most frequently associated with sport participation among girls. These desirable outcomes include the health benefits they may gain from participating in sports and whether they will enjoy participating. Enjoyment and potential health benefits have also been observed as motivational factors influencing self determination to engage in physical activity among adolescents [32]. Likewise, enjoyment of physical activity has been found to predict future time spent engaging in moderate to vigorous physical activity [33]. This may be explained by older adolescents’ increased autonomy to be physically active relative to younger adolescents whose physical activity may often be determined by external influences, such as parents and schools [34]. Other personal factors, such as sport self concept, physical self concept, or perceived competence, are also key predictors in future physical activity [33,34,35]. Thus, similar to findings in physical activity research, girls are most likely to continue participating in sports if they find them enjoyable and feel confident and competent when participating. Programs that allow girls to sample various sports may increase the opportunities for girls to discover which ones they enjoy and feel competent in.

The majority of studies included in this review lack theoretical framework; therefore, we aimed to categorize important variables associated with girls’ sport participation into the constructs of the theory of planned behavior to guide future research on this topic. First, variables categorized within the construct of attitudes included enjoyment, self concept, and expected outcomes of sport participation. Girls’ physical fitness, ability to improve their skills, and time spent playing the sport all influenced their intention to continue playing a sport [36,37,38]. The level of fun and enjoyment of sport has been associated with continued sport participation many times throughout the years [39,40,41,42]. Sport enjoyment has been directly related to teammate acceptance, which in turn has predicted sport commitment among adolescent girls [40]. When girls believe sport participation to be a way of developing friendships, they are more likely to take up and continue in sports [42,43,44]. Positive attitudes about being part of a team and competing are associated with sport participation [36,41]. Of the nine studies included in this review that were specifically analyzing motivation, seven identified variables that were subsequently operationalized as attitudes for the purposes of this review [36,39,41,42,43,44,45]. This finding highlights the importance of adolescent girls’ attitudes or behavior beliefs when considering their motivation to be involved in or continue sports over time.

The construct of perceived behavioral control encompasses many of these factors, including female adolescents’ perceptions of their competence and self worth, as well as their perceptions of available resources and opportunities to be involved in sports. Additionally, female adolescents’ self perceptions surrounding sport participation were often discussed with regard to body image [46,47,48,49]. Murphy et al. noted body image was commonly reported as a reason for uptake of a sport [46]. Some female athletes take up a new sport to improve body image, while others do not participate due to body image concerns [47,48]. Specific body-related concerns were related to breasts and thinness. Girls with concerns related to breasts bouncing during exercise or being embarrassed when changing due to breasts or bras were less likely to participate in sports [48]. Girls may think that society expects them to have thin bodies and that participating in certain sports may cause them to have a more athletic, muscular body that would make them less feminine [37]. Michaud et al. found female students often indicated a desire to lose weight when taking up new sports and were more likely to engage in dieting while participating in sports to achieve their weight loss goals [47]. The cultural norms communicated through media influence body image [50]. Present-day media tends to cover women’s sports by focusing on appearance over athletic performance [50]. Given the prevalence of social media and the implications it has on adolescent girls, further research should be conducted to understand the implications of social media on girls’ self perceptions and self esteem, and how those factors predict intentions to participate in sports. Interestingly, Dishman et al. found perceptions of appearance and body fat were related to physical self concept, a strong predictor of sport participation, but when they controlled for fitness and body mass index, appearance and body fat were unrelated to sport participation [51]. These findings are encouraging, as they indicate physical activity and sport participation positively influence self concept, independent of perceptions of physical appearance [51].

Perceived competence was also included in the construct of perceived behavioral control. Girls’ perception of athletic competence helps predict sport and exercise intention months and years later, because higher feelings of athletic competence encourage girls to maintain competence by continuing in a sport [52]. Girls who believe that they can handle the fitness load, skill level, and time spent playing a sport have an increased intention of sport continuation [46]. A study by Melman et al. reported significant stress among girls when attempting to manage the demands of schoolwork, extracurriculars, and leisure activities [53]. As girls grow older, they may feel more committed to their academic studies or take up other non-sport extracurricular activities [46,47,54,55]. This shift in commitments may decrease time spent training and their perceived competence in a given sport [12].

Certainly, socioeconomic status and other related aspects, including access to facilities, equipment, parental education level and employment status, town of residence, and perceived family wealth, were a common theme throughout the literature. We grouped these variables into the construct of perceived behavioral control, as girls may perceive their participation in sports as more or less attainable based on socioeconomic factors. Female adolescents with a lower socioeconomic status participated in sports at a lower rate than their peers with higher socioeconomic status [56,57,58,59,60,61,62]. Low socioeconomic status has been observed to be associated with girls’ poorer perceived outcomes of sport participation, lower perceived parental support, and greater barriers to participating in sports [56,57,62]. Multiple studies suggested increasing access to sports opportunities for girls with low socioeconomic status by improving or creating neighborhood recreational facilities, transportation to sporting venues, and awareness of local sports programs and opportunities [56,57]. Kanters et al. observed less socioeconomic status inequities among boys who attended middle schools with intramural sports programs as opposed to interscholastic sports; however, the increased access to sports opportunities gained through intramural programs did not positively influence participation among girls [58]. This gender difference may be due to the intramural sports programs’ use of mixed-gender sports, as middle school girls may be less inclined to participate in mixed-gender sports [30]. Improving adolescent girls’ access to sports opportunities through intramural sports programs may be more successful in promoting girls’ participation if they are not mixed gender. In addition to reducing barriers to sport participation among girls with low socioeconomic status, improving the parental support they receive may reduce the socioeconomic-related inequities in sport participation. Girls with low socioeconomic status receive lower parental support for sport participation [56], and this is especially true for older girls who tend to receive less sports-related praise and joint participation from parents compared to younger girls [63]. Efforts to improve parental support for sport participation may empower girls with greater perceived competence and self efficacy to capitalize on existing sporting opportunities or seek new opportunities.

Several variables were operationalized as subjective norms, or social influences, that impacted sport participation. These influences included both injunctive norms and descriptive norms. Injunctive norms, such as receiving approval or encouragement from family members and peers, were positively associated with girls’ sport participation [18,38,54,57]. Descriptive norms, which are formed by an individual’s perceptions of others’ behavior, were also important factors associated with girls’ sport participation. Girls were more likely to participate in sports if their parents provided encouragement [54,57], were sports participants [18,54,57,60], or regularly engaged in physical activity [64]. Eight of the articles identified parent, peer, or coach influence to be related to sport participation among girls. In one study, peer acceptance was found to mediate the association between self esteem and sport participation, with other important aspects of peer relationships, including support and encouragement, being noted as reasons girls take up and continue with sports [38,54,65]. Negative behaviors from coaches, including use of verbal and physical abuse, were associated with loss of motivation for girls to participate in sport [66]. With regard to parental factors, Eime et al. found family support to be the strongest and most consistent mediator between dimensions of socioeconomic status and sport participation [57]. That is, the degree to which girls perceive or receive support from family is the most important channel through which the effects of socioeconomic status influence sport participation [57]. Aside from socioeconomic status, in several of the articles, support from family was associated with continued physical activity and participation in team sports among adolescent girls [18,38,54,62,67,68,69].

This systematic review had several limitations. First, the records included in this review represented diverse study populations, employing a range of methodologies over a timespan of several decades, and evaluated participation in varying types of sports. We did not attempt to control for heterogeneity, which could impact our conclusions. Second, we limited the results to those records published in English, which could represent language bias. Third, we are limited by the constraints of our theoretical framework. There may be aspects of sport participation among girls that we may not have considered, as we focused our attention on operationalizing variables into constructs within the theory of planned behavior. Applying other theories may illuminate other variables that would provide a greater understanding of this phenomenon.

## 5. Conclusions

Through performing this systematic review, we sought to identify the factors associated with sport participation among girls and then group those factors into the theory of planned behavior constructs. Factors impacting girls’ sport participation were many, including personal, peer, family, socioeconomic, environmental, and other factors. Personal factors, such as self perceptions and desirable outcomes of sport (e.g., enjoyment and health benefits), were most frequently cited as factors influencing sport participation among girls. By operationalizing the identified factors, we provided a list of variables that may be tested to better understand which are most predictive of girls’ intentions to participate in sports. While the science surrounding sport and physical activity among adolescent girls has progressed tremendously over the last decade with respect to methodology and analytics, of the 36 records included in this review, only 7 were guided by theory, and all were quantitative studies. Future research would benefit from theory-driven prospective approaches to make clear and consistent predictions about factors impacting sport participation, as well as mixed-method approaches aimed to provide more robust understanding of girls’ experiences with and perceptions of factors impacting their participation in sports. Advancing the science with greater understanding of the multifactorial barriers and facilitators to sport involvement among girls will allow for implementation of appropriate and sustainable interventions aimed to support sport participation among girls across different populations and communities.

## Figures and Tables

**Figure 1 ijerph-19-03353-f001:**
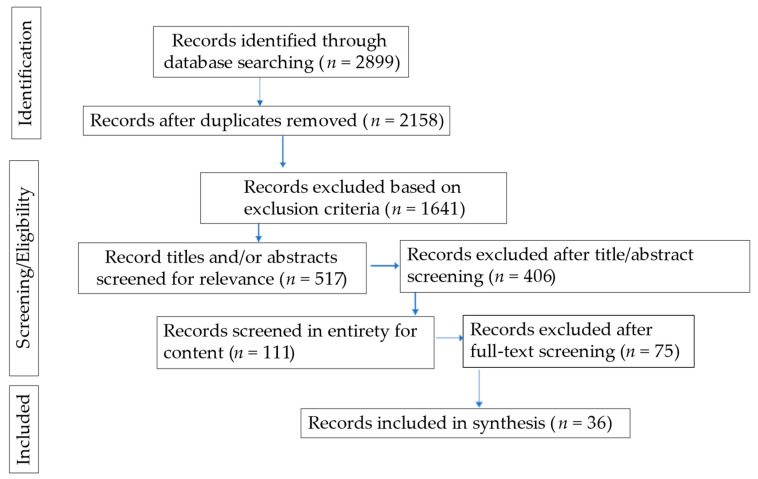
The PRISMA Group for reporting items for systematic reviews illustrating the record selection process.

**Figure 2 ijerph-19-03353-f002:**
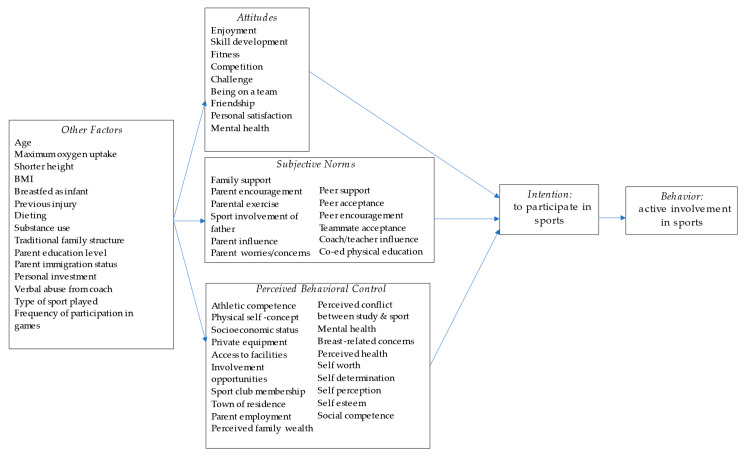
Adapted theory of planned behavior model with operationalized study variables.

**Table 1 ijerph-19-03353-t001:** Description of study characteristics.

Record	First Author (Year)	Topic/Title	Study Design	Population	*N*	Country
1	Agata (2018)	Body composition, fitness, and social correlates and sport participation	Secondary analysis	Males and females aged 14–15 years	238	South Africa
2	Atkins (2013)	Parental and peer influences on girls’ continuation in sports	Cross sectional survey	Females aged 10–14 years	227	United States
3	Balaguer (2012)	Self perceptions, self worth, and sport participation	Cross sectional survey	Males and females aged 11–16 years	917	Spain
4	Bedard (2020)	Sport participation and social competence	Secondary analysis	Males and females aged 9–14	2278	Canada
5	Cleland (2005)	Parental exercise association with child sport participation	Secondary analysis	Males and females aged 9–15	5929	Australia
6	Daniels (2006)	Sport participation, peer acceptance, and self esteem	Secondary analysis	Males and females aged 12–21	10,500	United States
7	Deflandre (2001)	Physical activity and sport involvement in high school students	Cross sectional survey	Males and females aged 16–19 years	48	France
8	DeJonge (2019)	Sport commitment and physical self-concept	Prospective longitudinal	Females aged 12–15 years	215	Canada
9	Delorme (2011)	Age and sport dropout in basketball players	Secondary analysis	Males and females aged 7–17 years	74,645	France
10	Dishman (2006)	Self-concept, self esteem, sport participation, and depression in girls	Cross sectional survey	Females aged 17–18	1250	United States
11	Dollman (2010)	Socioeconomic position and sport participation	Cross sectional study	Males and females aged 10–15 years	1737	Australia
12	Eime (2013)	Relationship between family support, access, socioeconomic status and sport participation	Cross sectional survey	Females aged 11–20 years	732	Australia
13	Engel (1994)	Gender role and stereotypes in women’s sports	Cross sectional survey	Females aged 12–16 years	200	England
14	Garn (2016)	Perceived teammate acceptance and sport commitment in adolescent female volleyball players	Cross sectional survey	Females aged 12–16 years	209	United States
15	Gill (1983)	Participation motivation	Cross sectional survey	Males and females aged 8–18 years	1138	United States
16	Guedes (2013)	Participation motivation	Cross sectional survey	Males and females aged 12–18 years	1517	Brazil
17	Guzman (2012)	Self-determination theory to predict sport dropout	Prospective longitudinal study	Males and females aged 11–19 years	857	Spain
18	Higginson (1985)	Socializing agents and female sport participation	Cross sectional survey	U13, junior high and senior high females	587	United States
19	Howie (2019)	Early life factors associated with trajectories of sport participation	Secondary analysis	Males and females aged 5–17	1679	Australia
20	Kanters (2013)	Impact of race, gender, and socioeconomics on sport participation	Cross sectional survey	Males and females aged 11–14 years	2582	United States
21	Longhurst (1986)	Motivation for participation in sports	Cross sectional survey	Males and females aged 8–18 years	404	Australia
22	Luiggi (2018)	Trends in sport participation	Cross sectional survey	Males and females aged 14–18 years	3218	France
23	McDonough (2005)	Friendship quality, self-concept and sport participation motivation	Cross sectional survey	Females aged 11–14 years	227	Canada
24	McMillian (2016)	Family structures and sport participation	Secondary analysis	Males and females aged 11–15 years	21,201	Canada
25	Michaud (2006)	Extracurricular sport participation among Swiss adolescents	Secondary analysis	Males and females aged 16–20 years	7428	Switzerland
26	Murphy (2017)	Impact of sport domain on future physical activity	Longitudinal	Males and females aged 10–18 years	873	Ireland
27	Saunders (2004)	Social variables and physical activity	Cross sectional survey	Females aged 13–14 years	4044	United States
28	Scurr (2016)	Influence of breasts on sport and exercise participation	Cross sectional survey	Females aged 11–18 years	2089	UK
29	Seabra (2008)	Socioeconomic variables and sport participation	Cross sectional survey	Males and females aged 10–18 years	3352	Portugal
30	Sit (2006)	Situational state balances and participation motivation	Cross sectional survey	Males and females aged 14–20 years	1235	Hong Kong
31	Snyder (1976)	Sport participation correlates among girls	Cross sectional survey	High school girls	500	United states
32	Tiggelman (2015)	Parental beliefs as determinants of sport participation in adolescents with asthma	Cohort study	Males and females aged 12–15 years	253	Netherlands
33	Toftegaard-Stockel (2011)	Factors associated with sport participation	Cross sectional survey	Males and females aged 12–16 years	6356	Denmark
34	Vella (2016)	Associations between sport participation and mental health	Secondary data analysis	Males and females aged 12 and 14 years	4023	Australia
35	Wattie (2014)	Age-related sport participation and dropout trends	Secondary data analysis	Males and females aged 11–20 years	3426	Germany
36	Yabe (2019)	Verbal abuse from coaches and sport motivation	Cross sectional study	Males and females aged 6–15 years	6791	Japan

**Table 2 ijerph-19-03353-t002:** Factors associated with sport participation among adolescent girls by data source.

Factor Categories/Study Variables	Total number of Data Sources	Data Source (Study Number Reported in Table 1)
**Personal Factors** Enjoyment Skill development Fitness Competition Athletic competence Physical self concept Personal satisfaction Social competence Self esteem Self perception Self determination Self worth Perceived health Perceived conflict between study and sport Dieting Substance use Personal investment Mental health Challenge Breast-related concerns	19	2, 14, 15, 30 15, 16, 21, 30 16, 21, 30 15, 21 3, 31 8, 10 17, 25 4 6 7 17 23 25 17 25 25 12 34 21 28
**Family Factors** Family support Parental exercise Traditional family structure Parent encouragement Sport involvement of father Parent influence Parent worries/concerns Perceived family wealth Parent education level Parent immigration status Parent employment	12	1, 7, 12, 27 5, 29, 33 24, 25 7, 32 7 18 19 24 25 33 33
**Biological/Physiological Factors** Age Maximum oxygen uptake Shorter height BMI Breastfed as infant Previous injury	9	7, 9, 13, 25, 33, 35, 36 7 19 34 19 18
**Peer Factors** Friendship Peer support Peer acceptance Peer encouragement Teammate acceptance Being on a team	9	16, 23, 30 1, 7 6 7 14 15
**Socioeconomic Factor** Socioeconomic status	6	11, 12, 14, 20, 22, 27
**Resources/Environmental Factors** Private equipment Access to facilities Co-ed physical education Involvement opportunities Sport club membership Town of residence	6	7 12 13 14 25 33
**Sport/Coach Factors** Coach/teacher influence Verbal abuse from coach Type of sport played Frequency of participation in games	4	18 34 26 34

## Data Availability

Not applicable.

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
