# Peer review of "A Systematic Review of Factors Associated with Sport Participation among Adolescent Females"

_ijerph, 2022, doi:10.3390/ijerph19063353_

Round 1

Reviewer 1 Report

My recommendations are as follows
Introduction of bibliographic indexes according to the editing rules.
In the abstract, mention the factors that are personal in both categories, it is not clear, I recommend clarification.
Line 106 you wrote and in capital letters, I recommend the correction.
Lines 155-156 repeat the idea, lines 105-106, I recommend deleting.
Sections 3.1,3.2,3.3,3.4 repeat the ideas presented in figure 2, I recommend deleting or rewriting.
I recommend correcting the bibliography according to the rules of the journal.
I recommend that the discussion section be restructured on the main categories of factors.
The first conclusion is also mentioned in lines 76-77, I recommend clarification.
Lines 360-362 is not a conclusion.
I recommend rewriting the conclusions section with a focus on the concrete results of this review.

Author Response

Our thanks to you for your excellent review of our manuscript. The attached table addresses your comments and critique including our response and the line numbers of the revisions within the manuscript.

Reviewer 2 Report

Title

The title of the manuscript reads well and is an accurate presentation of the manuscript content.

Abstract

Overall, the abstract is well written. My only concern is that Authors should mention what are the recommended levels of physical activity in this context.

Introduction

Overall, the introduction reads well. My only and the biggest concern is that the introduction is too short. The Authors should elaborate on important topics and terms relevant to research of physical activity. For example, several research papers include many common key terms that are highly relevant to physical activity – for example motivation. I think motivation is one key concept that affects adolescents’ physical activity. In addition, recent research has highlighted that it is specifically intrinsic motivation that is related to adolescents’ daily physical activity (Kalajas-Tilga et al., 2020).

Kalajas-Tilga, H., Koka, A., Hein, V., Tilga, H., & Raudsepp, L. (2020). Motivational processes in physical education and objectively measured physical activity among adolescents. Journal of Sport and Health Science, 9(5), 462–471. https://doi.org/10.1016/j.jshs.2019.06.001

There are also several studies which demonstrate that intrinsic (or autonomous) motivation is the key predictor of the theory of planned behavior constructs (in the context of adolescents’ physical activity), for example see this study by:

Polet, J., Schneider, J., Hassandra, M., Lintunen, T., Laukkanen, A., Hankonen, N., Hirvensalo, M., Tammelin, T. H., Hamilton, K., & Hagger, M. S. (2021). Predictors of school students’ leisure-time physical activity: An extended trans-contextual model using Bayesian path analysis. PLOS ONE, 16(11), e0258829. https://doi.org/10.1371/journal.pone.0258829

Also, after the introduction and before methods Authors could add paragraph “The Present Study”.

Method

2.1. Theoretical Framework - I believe much information from here could be moved to the introduction section. This paragraph is not so relevant to the Methods section. Please revise.

Figure 1 – based on the figure it is not that sure where the box “records after duplicates removed” should go? Identification or screening?

Figure 2 – it is hard to read this figure, could Authors increase a bit the size of these words?

Results

The results are clear, and everything is well explained to the reader.

Discussion

The results are discussed from multiple angles and the Authors have put the results in proper context. My only concern is that I would like to see more discussion about the motivation related to physical activity or sport participation.

Author Response

Our thanks to you for your excellent review of our manuscript. Please see the attached table addressing your comments and critique including our response and the line numbers of the revisions within the manuscript.

Reviewer 3 Report

Thanks for the opportunity to read this article.

After reading, I have serious concerns that I present below.

Line 31. When you use the word sport, do you only refer to PA practice in the sporting context?

Line 46. After the semicolon, you must add: WHO. (2020). WHO guidelines on physical activity and sedentary behaviour. World Health Organization. https://www.who.int/publications/i/item/9789240015128

Lines 47 and 51. This reference must be dropped and can be added:

Guthold, R., Stevens, G. A., Riley, L. M., & Bull, F. C. (2020). Global trends in insufficient physical activity among adolescents: a pooled analysis of 298 population-based surveys with 1.6 million participants. Lancet Child Adolesc Health, 4(1), 23-35. https://doi.org/10.1016/S2352-4642(19)30323-2

Marques, A., Henriques-Neto, D., Peralta, M., Martins, J., Demetriou, Y., Schonbach, D. M. I., & Matos, M. G. (2020). Prevalence of physical activity among adolescents from 105 low, middle, and high-income countries. International Journal of Environmental Research and Public Health, 17(9). https://doi.org/10.3390/ijerph17093145

Lines 51-55. I didn’t understand the injury part. Does this have anything to do with the purpose of the work?

Line 76-99. All this part can be eliminated. The addition of examples to explain the author’s idea is strange. The reader wants to know information about the review protocol in a systematic review.

Line 101. The article was submitted in 2022, but the search was performed in 2020. There is a very large time gap. This survey should have been updated.

Lines 104-106. Keywords seem not to be enough to include all articles on the topic.

For example, for the term sport, "exercise", "physical activity", "sports club" should also have been used.

When presenting the results, I did not understand the relationship established with TPB.

Author Response

(The authors gave the same response as above.)

Round 2

Reviewer 1 Report

no comments

Reviewer 2 Report

Authors have done well job by revising the manuscript.

Reviewer 3 Report

With the revisions, the authors substantially improved the quality of the article.